# Phytocannabinoids Reduce Inflammation of Primed Macrophages and Enteric Glial Cells: An In Vitro Study

**DOI:** 10.3390/ijms241914628

**Published:** 2023-09-27

**Authors:** Gal Cohen, Ofer Gover, Betty Schwartz

**Affiliations:** Department of Biochemistry Food Science and Nutrition, The Robert H. Smith Faculty of Agriculture, Food and Environment, The Hebrew University of Jerusalem, Herzl Street 229, Rehovot 7610001, Israel; galgalita124@gmail.com (G.C.); ofer.gover@mail.huji.ac.il (O.G.)

**Keywords:** phytocannabinoids, J774A1 M1 macrophages, enteric glial cells

## Abstract

Intestinal inflammation is mediated by a subset of cells populating the intestine, such as enteric glial cells (EGC) and macrophages. Different studies indicate that phytocannabinoids could play a possible role in the treatment of inflammatory bowel disease (IBD) by relieving the symptoms involved in the disease. Phytocannabinoids act through the endocannabinoid system, which is distributed throughout the mammalian body in the cells of the immune system and in the intestinal cells. Our in vitro study analyzed the putative anti-inflammatory effect of nine selected pure cannabinoids in J774A1 macrophage cells and EGCs triggered to undergo inflammation with lipopolysaccharide (LPS). The anti-inflammatory effect of several phytocannabinoids was measured by their ability to reduce TNFα transcription and translation in J774A1 macrophages and to diminish S100B and GFAP secretion and transcription in EGCs. Our results demonstrate that THC at the lower concentrations tested exerted the most effective anti-inflammatory effect in both J774A1 macrophages and EGCs compared to the other phytocannabinoids tested herein. We then performed RNA-seq analysis of EGCs exposed to LPS in the presence or absence of THC or THC-COOH. Transcriptomic analysis of these EGCs revealed 23 differentially expressed genes (DEG) compared to the treatment with only LPS. Pretreatment with THC resulted in 26 DEG, and pretreatment with THC-COOH resulted in 25 DEG. To evaluate which biological pathways were affected by the different phytocannabinoid treatments, we used the Ingenuity platform. We show that THC treatment affects the mTOR and RAR signaling pathway, while THC-COOH mainly affects the IL6 signaling pathway.

## 1. Introduction

Intestinal inflammation is mediated by a subset of cells populating the intestine [1,2]. Macrophages are key players in the innate immune system, and are involved in phagocytosis, antigen presentation, and secretion of various cytokines, chemokines, and growth factors that protect the body from infection [3]. In the integral part of the normal intestinal tissues, macrophages are well-established in the lamina propria and in Peyer’s patches, where they function as immune effector cells against any pathogenic attack [4]. In the presence of inflammatory stimuli, macrophages polarize toward the pro-inflammatory M1 phenotype that produces high levels of inflammatory cytokines and chemokines in order to eliminate pathogens [4]. The exposure of macrophages to lipopolysaccharide (LPS) or Tumor necrosis factor alpha (TNF-α) induced their differentiation to the M1 phenotype [5]. Typically, M1 macrophages secrete toxic cytokines, such as TNF-α, IL-1β, IL-6, IL-12, IL-18, IL-23, nitric oxide, and reactive oxygen species (ROS). In addition, M1 macrophages decrease the stimulation of the anti-inflammatory cytokine IL-10 [6]. In contrast, wound healing environment promotes macrophage polarization to the anti-inflammatory M2 state with increased production of anti-inflammatory cytokines, leading to alleviating inflammation, tissue repair, and remodeling [7,8]. Emerging evidence has shown that recruitment of a large number of M1 inflammatory macrophages to inflamed tissues associated with high production of pro-inflammatory cytokines contributes to inflammation and tissue damage in inflammatory diseases [9]. Macrophages exhibit a particularly vigorous response to LPS, a surface component of most Gram-negative bacteria, and TNF-α in response to inflammation or injury. The dysfunction of the mucosal immune response in IBD is characterized by abnormalities in both the innate and adaptive immune systems [10]. The final common pathway of this dysregulated immune activation is abundant infiltration of immune cells, such as macrophages and monocytes, in the intestinal mucosa [11,12,13].

The gastrointestinal tract differs from all other peripheral organs in that it includes an extensive enteric nervous system (ENS), differing from all other peripheral organs. The ENS is characterized by the presence of neurons and enteric glial cells (EGCs), which are arranged into interconnected ganglia distributed between the plexuses. There is evidence indicating that inflammatory diseases of the gut are characterized by changes affecting enteric glial cells [14,15]. EGCs are activated by exogenous stimuli that lead to over-release of neurotrophins, growth factors, and cytokines that, in turn, recruit infiltrating immune cells such as macrophages, neutrophils, and mast cells into the colonic mucosa [16,17,18]. In fact, EGCs exert a key role in the maintenance of gut homeostasis through cooperating with surrounding cells. Specifically, EGCs assure the correct trophism of neurons in the ENS [1,18], protect neurons from oxidative stress [19], control epithelial barrier functions by reducing epithelial permeability, and actively participate in the course of intestinal inflammation acting as the first defensive line of the ENS [1,17]. Enteric glial cells have also gained a particular interest for IBD pathogenic processes since they morphologically and functionality resemble astrocytes, which maintain homeostasis in the central nervous system [20]. The response to different insults, such as inflammation and infection [21,22], is manifested by expressing glial fibrillary acidic protein (GFAP) and S100 calcium-binding protein B (S100B) [10,23]. In the disease state, inflammation can convert EGCs to a “reactive EGC phenotype” characterized by over-release of neurotrophins, growth factors, and cytokines that, in turn, recruit infiltrating immune cells such as macrophages, neutrophils, and mast cells in the colonic mucosa [17,24]. In this pathological condition, increased GFAP and S100B protein expression from EGCs have also been observed. These two proteins provide reliable biomarkers of glial activation in the intestinal tissue [23,25]. 

The endocannabinoid system (ECS) comprises endogenous cannabinoids (endocannabinoids (eCBs)), cannabinoid receptors, and proteins that transport, synthesize, and degrade eCBs. Most components of the ECS are multifunctional. Thus, rather than being a discrete system, the ECS influences, and is influenced by, many other signaling pathways. This is especially important to consider when assessing the effects of ECS-targeting drugs [26]. The actions of most phytocannabinoids is mediated via receptors that are part of the ECS, through agonistic and antagonistic actions at specific receptors sites [26,27] and in varying degree of affinity [28]. The best-known receptors of the ECS are cannabinoid receptor 1 (CB1R) and cannabinoid receptor 2 (CB2R). Both receptors are G-protein coupled receptors (GPCR) that activate intracellular signaling [29]. For example, the mitogen-activated protein kinase (MAPK) pathway results from G-protein-coupled receptors activation [30], such as the resulting CB1 stimulation [27,31]. CB1Rs are found mainly on neurons, in the brain, and in the spinal cord, and they are expressed by some astrocytes [32,33]. In addition, CB1Rs are expressed in many peripheral organs and tissues, including in the gastrointestinal tract [34,35], in the enteric nervous system [36], in the healthy colonic epithelium, in the gut smooth muscles, and in the submucosal myenteric plexus [37].

CB2 receptors are primarily expressed in cells of immune origin [38], including microglia [39], though they may also be expressed in neurons [39], particularly in pathological statuses [33]. Indeed, there is markedly more mRNA for CB2R than CB1R in the immune system [38]. CB2R are also present in epithelial and immune cells from the gastrointestinal tract [40] and, in contrast to CB1 receptors, CB2 receptors are highly expressed in macrophages and in colonic epithelium tissue taken from IBD patients [37]. In fact, increased epithelial CB2 receptor expression in human inflammatory bowel disease tissue implies an immunomodulatory role that may affect mucosal immunity. The endocannabinoid system has been demonstrated to be activated in several conditions, including inflamed intestine in mice, and, thus, expresses an increased amount of endocannabinoid receptors [41].

Our aim in this study was to analyze the putative anti-inflammatory effect of nine selected pure cannabinoids in vitro in J774A1 macrophages cells and enteric glial cells (EGCs) triggered to undergo inflammation with lipopolysaccharide (LPS). The anti-inflammatory effect of the phytocannabinoids was measured by their ability to reduce TNFα transcription and translation in J774A1 macrophages and to diminish S100B and GFAP secretion and transcription in EGCs.

## 2. Results

### 2.1. Effects of Phytocanabinoids on Inflammation in Murine Macrophages and EGC

The detection of TNFα levels provided us with an acceptable criterion to evaluate the extent of inflammation. The maximal response to LPS elicitation of J77A1 cells was detected by the level of secretion of TNFα and measured after treatment with LPS (0.05 µg/mL) for 4 h (Appendix A). Treatment with dexamethasone at a concentration of 5 µM (Appendix A) provided us the most suitable control indicator of reduction of inflammatory response in J77A1 cells. All further analyses were performed with 0.05 µg/mL of LPS for 4 h for secretion of TNFα and 0.05 µg/mL of LPS for 1 h for gene expression (Appendix A).

EGCs play a fundamental role in gut maintenance and inflammation. During intestinal inflammation, EGCs enter reactive gliosis and overexpress S100B protein, a molecule that plays a pivotal role in the downstream signaling process of EGC inflammation [16]. Furthermore, it has been shown that EGCs express CB2 receptors [42]. Quantification of cellular S100B protein was performed by ELISA. Sparstolonin B (Ssnb) is a polyphenol which inhibits TLR4 activation by blocking the binding of TLR4 to MyD88 (an important mediator of almost all the TLR downstream singling), thereby suppressing Nuclear factor-κB (NF-κB) [43,44]. Treatment with Ssnb at a concentration of 10 µg/L provided us the most suitable control indicator of reduction of inflammatory response in EGCs. Afterwards, EGCs were incubated with 1 μg/mL LPS for 24 h. EGCs express not only S100B, but also express high levels GFAP [45] upon inflammatory stimulation. 

All cannabinoids were assayed for cytotoxicity using the MTT method [46] on J774A1, and concentrations that did not reduce cell viability were selected for further analysis (Appendix A). For CBGA and CBDA, all concentrations tested reduced cell viability by at least 50% and, hence, were not evaluated further (Appendix A).

#### 2.1.1. THC

The putative cytotoxic effect of Delta-9-tetrahydrocannabinol (THC) was evaluated by (3-(4,5-dimethylthiazol-2-yl)-2,5-diphenyltetrazolium bromide) tetrazolium reduction assay (MTT). We demonstrate that THC was not toxic to the cells at concentrations up to 10 µg/mL (Appendix A). THC induced the greatest effect on reducing TNFα secretion at either 0.1 or 0.05 µg/mL (Figure 1A). This reduction was comparable to the effect of dexamethasone. Above these concentrations, THC increased TNFα secretion to levels equal to (0.1–1 µg/mL) or above (5–10 µg/mL) treatment with LPS (Figure 1A). The extent of reduction of secretion did not mirror the effects measured on gene expression (Appendix A).

Pretreatment of EGCs with 0.05 μg/mL and 0.1 μg/mL THC led to a significant reduction in inflammation by lowering S100B protein levels (Figure 1B) and GFAP expression levels (Figure 1C). In EGC, a dose-dependent reduction in inflammatory markers was observed. 

#### 2.1.2. CBD

Cannabidiol (CBD) is the second most abundant phytocannabinoid with non-psychoactive effects, which makes it well tolerated by consumers compared to THC [47]. CBD was not toxic to the cells up to 3 µg/mL (Appendix A). Treatment J77A1 macrophages with CBD concentrations under the cytotoxicity level induced increased secretion of TNFα (Figure 1D). Furthermore, the combination of THC and CBD abolished THC’s anti-inflammatory effect (Figure 1D).

#### 2.1.3. THC-COOH

Metabolism of THC occurs mainly in the liver by microsomal hydroxylation and oxidation catalyzed by enzymes of the cytochrome P450 (CYP) complex [48]. The first product is 11-Hydroxy$\Delta^{9}$-tetrahydrocannabinol (11-OH-THC), while 11-Nor-9-carboxy-$\Delta^{9}$-tetrahydrocannabinol (11-THC-COOH or THC-COOH) is the final product. THC-COOH is not psychoactive and possesses anti-inflammatory and analgesic properties by mechanisms similar to those of nonsteroidal anti-inflammatory drugs [49]. Concentrations of THC-COOH above 10 µM were toxic to J774A1 cells (Appendix A) and, above 100 µM, to EGC (Appendix A). Most of the concentrations of THC-COOH (1 nM, 10 nM, 100 nM, 1 μM, 2 μM, 5 μM, and 10 μM) that we tested did not exert a significant effect on TNF-α secretion. However, 500 nM increased TNF-α in those cells (Figure 1F). Concentrations from 1 nm to 10 μM significantly reduced S100B secretion and expression (Figure 1G and Appendix A) and GFAP expression (Figure 1H). 

#### 2.1.4. THCA

In the plant trichrome, THC is stored in its acidic form, ∆^9^-tetrahydrocannabinol acid (THCA) [50]. THCA is devoid of psychotropic effects [51]. THCA needs to undergo decarboxylation to THC to produce psychotic effects; this decarboxylation is spontaneous and requires heat [27]. We show in cell viability experiments that THCA is not cytotoxic to J774A1 or to EGC up to 10 µM (Appendix A and Appendix A). In concentrations below 10 µM, THCA increased secretion of TNFα in J774A1 cells compared to 0.05 µg/mL LPS without pretreatment (Figure 2A). In concentrations of 0.1–10 µM, THCA reduced S100B secretion and expression (Figure 2B and Appendix A) and GFAP expression (Figure 2C). THCA alone did not elevate S100B secretion or GFAP expression (Figure 2B,C). 

#### 2.1.5. CBC and CBG

Pretreatment with CBC and CBG did not exert any cytotoxic effect on J77A1 cells below 10 µM and 20 µM, respectively (Appendix A). CBC treatment of J774A1 macrophages showed a U-shaped curve where 1 nM, 10 µM, and 20 µM significantly elevated TNFα secretion compared to LPS control, and concentrations between 10 nM and 1 µM were insignificant related to control (Figure 2D). Pretreatment with 10 nM–1 µM CBC significantly reduced S100B secretion and expression to the same levels as Ssnb (Figure 2E and Appendix A). Concentrations of 10 nM, 100 nM, 500 nM, and 1 μM CBC did not exert any effect on GFAP expression (Figure 2F). 

CBG did not exert any significant effect on TNF-α secretion in comparison to the positive control (0.05 μg/mL LPS) over the whole range of CBG concentrations tested (0.05–20 μM CBG) (Figure 2G).

CBG treatment induced an anti-inflammatory effect in EGCs at 1–10 µM for S100B secretion (Figure 2H). However, the effect on secretion differs from the effect on relative expression of S100B (Figure 2H and Appendix A). A discrepancy between S100B expression and translation in EGCs after treatment with 20 μM CBG (Figure 2H and Appendix A) was seen where it increased S100B secretion yet showed decreased transcription. Only 1 µM of CBG significantly reduced GFAP expression, whereas a dose-dependent increase in GFAP expression was seen with and without LPS (Figure 2I). 

#### 2.1.6. THCV

Δ9-Tetrahydrocannabivarin (THCV) is described as a phytocannabinoid belonging to one of the minor phytocannabinoids. The name “minor phytocannabinoids” has been used to define phytocannabinoids different from Δ^9^-THC, CBD, CBG, and CBC. THCV decreases signs of inflammation and pain in an acute inflammation model in mice partly via CB1 and/or CB2 receptor activation [52]. We show, for J77A1 macrophages, that THCV is not cytotoxic below 15 nM (Appendix A). THCV treatment of J774A1 cells showed significant increases in TNFα secretion from cells at concentrations of 0.2 nM, 0.5 nM, 1 nM, and 3 nM compared to the 0.05 μg/mL LPS group (Figure 3). Although it is not significant, 5 nM, 7 nM, 10 nM, and 15 nM showed an increase in TNF-α secretion compared to J774A1 cells treated with 0.05 μg/mL LPS (Figure 3).

### 2.2. RNA Sequencing Analysis

EGCs were treated for 1 h with 0.1 µg/mL of THC or 10 nM of THC-COOH, after which 1 µg/mL of LPS was added for 4 h. Additionally, a positive control of 1 µg/mL LPS and negative control cells were used. Transcriptomic analysis revealed 23 differentially expressed genes (DEG) (fold change >1.6, *p*v < 0.05) in the control vs LPS (10 upregulated and 13 downregulated), 26 DEG THC + LPS vs LPS (10 upregulated and 16 downregulated) and 25 DEGs when comparing THC-COOH + LPS vs LPS (16 upregulated and 9 downregulated) (Figure 4). Pretreatment with THC before LPS resulted in downregulation of the apoptosis related genes *mob1a* (*p*v < 0.05, −1.7FC) [53] and *ptma* (*p*v < 0.05, −2FC). THC downregulated *adm* (*p*v < 0.05, −1.6FC), a gene that is upregulated in inflamed neurons, and antagonists to *adm* inhibit the release of nNOS and macrophage recruitment [54]. *Ap1S3* was upregulated by pretreatment with THC (*p*v < 0.05, 3.5FC). Knockout of *ap1s3* in keratinocytes results in upregulation of IL1 and TNFα [55]. IL1 is a strong activator of IBD [56,57]. Oxidative stress genes were also affected by THC with downregulation of *gpx8* (*p*v < 0.05, −2FC) and upregulation of *oplah* (*p*v < 0.05, 3FC). 

THC-COOH is the final metabolite of THC metabolism in the liver. Pretreatment of THC-COOH resulted in upregulated genes relating to cellular metabolism, including *cox6c*, *cox7b*, and *gbe1* (Figure 4). THC-COOH upregulated *psma2*, a component of the 20S subunit. Knockdown of this gene results in reduced immune response in human lung cells [58]. 

#### Pathway Enrichment Analysis

To evaluate the biological pathways affected by the different pretreatments, we used the IPA platform (*p*v < 0.05). Addition of LPS affected 34 canonical pathways (−log(*p*-value > 1) (Figure 5). The 4 major pathways (−log(*p*-value) 2.88–3.78) were related to: oxidative phosphorylation (−log(*p*-value) 3.78), mitochondrial dysfunction (−log(*p*-value) 3.51), ElF2 signaling (−log(*p*-value) 2.88), and HER-2 signaling (−log(*p*-value) 2.88). Downregulation of *cox6c* (log2FoldChange −2.46) and *cox7b* (log2FoldChange −2.63) and upregulation of *mt-co1* (log2FoldChange 1.97) results in changes in oxidative phosphorylation, mitochondrial dysfunction, and HER-2 signaling. Pretreatment with THC for 1 h also affected the Elf2 signaling pathway through downregulation of *gm15489*, *mt-rnr1*, and *rpl2*1; however, *rpl7a* was upregulated. The most affected pathway by pretreatment with THC was the mTOR pathway ((−log*p*v 3.96) Figure 5). This manifested through downregulation of four genes (*ddit4*, *gm15483*, *mt-rnr1*, *rps6kc1*). The only pathway to show mild downregulation was the retinoic acid receptor (RAR) (zScore −2). Interestingly, the final metabolite of THC, THC-COOH, affected different pathways when compared to THC. The carboxylated metabolite elicited changes in IL-6 signaling (*tab1* −logFC−2.2, *tnfaip6* −logFC2.4) and iNos signaling (Tab1 −logFC−2.2); both are canonical pathways related to inflammation. In general, carboxylated THC affected fewer pathways compared to THC above the threshold of 1.3 (Figure 5B,C). IPA analysis identifies genetic networks that are affected by the DEG regardless of the direction of the expression change. A total of 0.1 µg/mL LPS affected the cell cycle and cell death network through 11 DEGs (Figure 5D). Pretreatment with THC affected the 17 DEGs of the proteosome network (Figure 5E). The main network to be affected by pretreatment of THC-COOH was related to cell death and survival (Figure 5F).

## 3. Discussion

In this study, we aimed to perform an in-depth assessment of the putative anti-inflammatory effects of both major and minor phytocannabinoids on macrophages and enteric glial cells. Macrophages are resident cells of almost every tissue in the body and provide key orchestrators of chronic inflammatory disorders. Macrophages have been reported to play a role in the pathological progression of UC (Ulcerative Colitis) disease in comparison with other leukocytes [59]. The glial cells in the gut represent the morphological and functional equivalent of astrocytes and microglia in the central nervous system, and they play essential roles as regulators of intestinal homeostasis [60,61]. Although no CBr have been detected in EGCs, there is mounting evidence on the effects of modulation of EGCs by the ECS [62]. During intestinal inflammatory reaction, EGCs release glial markers such as S100B and GFAP. Altered expression of S100B and GFAP has been reported in several intestinal inflammatory disorders in humans, such as inflammatory bowel disease [16,45], celiac disease [17], and postoperative colitis [40]. However, EGCs were not in the focus of cannabinoid research [62].

Our results show that THC significantly reduced TNFα secretion preferentially at low concentrations 0.05–0.5 µg/mL, whereas, above 0.55 µg/mL, TNFα secretion was increased. At 5–105 µg/mL, secretion was above the effect of 0.5 µg/mL LPS, showing an additive effect of THC on TNFα secretion. This is in accordance with the known biphasic effect of THC [63,64]. Becker et al. showed a reduction in in cytotoxic T cells and T-bet + TH1 cells in THC and THC + CBD mice subjected to dextran sodium sulfate (DSS) and 2,4,6-trinitrobenzenesulphonic acid (TNBS) induced UC [65] compared to vehicle control and CBD. Yakhtin et al. showed that activated peritoneal macrophages treated with THC and THC extract have reduced NO, IL6, TNFα, CXCL2, and G-CSF compared to vehicle control [66]. These data point to the advantage of using low or even ultralow doses of THC, in accordance with the reversal of cognitive impairment in old mice and spatial memory tests in old female mice [67,68]. Transcription of TNFα was not affected by pretreatment with THC; this was evident from RTqPCR (Appendix A) and transcriptomics. This is inline with other reports showing the instability of TNFα mRNA vs protein [69,70]. Pretreatment of EGC with low concentrations of THC markedly reduced S100B secretion and expression as well as GFAP expression (Figure 1B,C). Taken together, these results show the significant immunosuppressive effect exerted by low doses of THC in our in vitro model.

We demonstrate herein that CBD augmented the secretion of TNFα in all concentrations tested except for 3 µg/mL, where it was the same as 0.5 µg/mL LPS control (Figure 1D). CBD has been used for the treatment of inflammation and other comorbidities [71,72]. Our results may be due to the low concentrations used, as it has been shown for T cells [73]; yet, it is notable that the concentrations we used were optimized to be non-cytotoxic. Furthermore, CBD abolished THC reduction of TNFα secretion (Figure 1E). Some data suggest that CBD can indirectly modulate THC via CB receptors [74] or by being an allosteric modulator that alters the efficacy of orthostatic ligands [75,76]. Filippes et al. showed reduced S100B secretion from LPS-treated mice pretreated with CBD; this reduction could be due to a systemic response in which CBD does not directly interact with EGC, like in our model [71].

Upon consumption, THC is metabolized in the liver to 11-OH-THC and then to the inactive metabolite THC-COOH [77]. THC-COOH did not reduce inflammatory markers in murine macrophages (Figure 1F); yet, it reduced all reactive glycolysis markers in EGC (Figure 1G,H and Appendix A). In the plant trichomes, THC is stored in its acidic form THCA [78]; THCA is not psychoactive, and, upon heating (smoking or baking), it undergoes a decarboxylation reaction and is transformed to THC [27]. At 1–70 nM, THCA significantly increased TNFα secretion from J774A1 macrophages. At higher concentrations, 100 nm–1 µM, no significant inflammatory effect was observed when compared to positive LPS control; yet, no reduction of TNFα secretion was measured for any of the concentrations tested (Figure 2A). In EGCs, THCA reduced both S100B secretion and expression, as well as GFAP expression at all concentrations tested (Figure 2B,C and Appendix A). This indicates that THCA can prevent induction of reactive gliosis in EGCs. C. sativa flower extract fraction, rich in THCA, reduced IL-8 secretion, a marker for IBD inflammation, from HCT116 colonocytes, and CaCO2 cells using GPR55 antagonists blocked the reduction in IL-8, indicating that THCA exerts its effect through this receptor [79].

THCV is a minor cannabinoid from C. sativa with evidence of medicinal properties in metabolism [80], nausea [81], obesity and insulin sensitivity [35], pain [52], and inflammation [82]. Low concentrations, 0.2–3 nM, of THCV elevated murine macrophage TNFα secretion, whereas higher concentrations were not different from LPS-treated cells (Figure 4B). Overall, the results show that THCV did not improve inflammation markers in our system. This is in accordance with results of Rao et al. [82] that showed that THCV did not reduce LPS-induced NO production in RAW264.7 and similarly to the observed for keratinocytes [83].

CBC is considered one of the main four cannabinoids in the *Cannabis sativa* plant. It has been shown to have therapeutic properties through activation of TRPA1 and inhibition of degradation of cannabinoids [84,85,86]. An increase in TNFα secretion was generally observed in J774A1 macrophages. The lowest and the highest concentrations of CBC generally increased TNFα secretion; the most significant effect was achieved with 1 nM and 10–20 µM compared to LPS control (Figure 2D). Concentrations of 10 nM to 1 µM significantly reduced S100B secretion and expression in EGC (Figure 2E and Appendix A respectively). However, GFAP mRNA expression was not changed by pretreatment with CBC. Cumulatively, these results show that CBC does not reduce inflammation on the tested primed cells according to the results of all of the markers that we measured (Figure 2F).

CBG may exert therapeutic effects through modulation of transient receptor potential (TRP) channels, cyclooxygenase (COX-1 and COX-2) enzymes, and cannabinoid 5-HT1A and α2 adrenergic receptors [84,87,88,89]. Non-cytotoxic concentrations of CBG (0.05–20 µM) did not reduce TNFα secretion (Figure 2G). At 1–10 µM, CBG reduced S100B secretion by EGC, whereas, at 20 µM, S100B secretion was elevated with and without LPS (Figure 2H). Expression levels of S100B were reduced in all tested concentrations (Appendix A). Expression of GFAP was only reduced at 1 µM CBG and was significantly increased at 20 µM with or without LPS (Figure 2I). Collectively, it can be stated that CBG is effective on EGC at concentrations of 1–10 µM.

To evaluate a more systematic cellular response, transcriptomic analysis was conducted on control untreated EGCs compared to EGCs treated with LPS 1 µg/L for 24 h, pretreated for 1 h with 0.1 µg/mL THC or 10 nM THC-COOH, and then with 1 µg/L LPS (see Materials and Methods). Incubation with LPS-induced 23 DEGs (fold change > 1.6, *p*v < 0.05) (Figure 4A). Pathway analysis revealed that the main pathways affected by LPS were oxidative phosphorylation, mitochondrial dysfunction, and EIF2 signaling (Figure 5A). Oxidative phosphorylation and mitochondrial dysfunction were affected by downregulation of *cox6c*, *cox7b*, and *mt-co-1*, which are all part of the mitochondrial respiratory complex. It has been shown that the mitochondria are active in infection and inflammation through the release of cytokines and the activation of the inflammasomes [90]. Both *cox6c* and *cox7b* are part Cytochrome c Oxidase, the terminal enzyme of the mitochondrial respiratory chain. The reduction in oxidative phosphorylation agrees with previous reports showing a shift from oxidative phosphorylation to glycolysis in LPS-induced glial cells [91,92]. Elf2 was shown to be activated via phosphorylation in RAW 264.7 cells by *Yersinia pseudotubercu* infection [93], causing a reduction of protein synthesis by negatively affecting the exchange of GDP to GTP in the β-subunit of eLF2. In BV-2 microglial cells, LPS can cause excessive mitochondrial fission and ROS generation [94]. LPS has been shown to elevate oxidative stress in BV-2 microglial cells and in the brain, as well as in other organs [95,96]. EIF2 phosphorylation is increased in murine macrophages that are exposed to bacterial infection, causing a reduction in protein synthesis [93]. IPA analysis identifies genetic networks that are affected by the DEG regardless of the direction of the expression change. Treatment with LPS affected developmental disorder, hereditary disorder, and metabolic disease networks (12 DEG), along with cell cycle, cell death and survival, organismal injury, and abnormalities (11 DEG) (Figure 5D). This is in accordance with Juknat et al. [97].

Pretreatment for 1 h with 0.1 µg/mL THC before incubation with LPS resulted in 26 DEGs (10 upregulated and 16 downregulated) compared to no pre-incubation. The main pathways affected were mTOR signaling, EIF2, and retinoic acid receptor (RAR) activation, which was downregulated (z Score −2). mTOR was affected by downregulation of *ddit4*, *gm15483*, *mt-rnr1*, and *rps6kc1*. mTOR signaling has been implicated in inflammation processes. Mammalian target of rapamycin (mTor) is a conserved serine/threonine protein kinase belonging to the phosphoinositide 3-kinase (PI3K) family. It has been shown in CNS microglial cells that LPS activates mTor activity resulting in nitric oxide (NO) and prostaglandin E2 and D2 [98]. Inhibition of mTor using rapamycin inhibited these effects through reduction of COX2 and NOS2 [99]. RAR is essential for enteric nervous system (ENS) development. Knockout of RAR led to reduction of submucosal neurons yet did not reduce enteric glia cells primed by SOX10 [100]. Retinoic acid ameliorates IBD through NFƙB signaling in the colitis model, along with RAW264.7 macrophages [101]. Most studies indicate activation of RAR in inflammation [100,101] yet our data indicate a reduction of the RAR pathway in glial cells pretreated with THC. Network analysis revealed that THC influenced the proteosome network (17 DEGs). It has been previously shown that mTOR regulates protein synthesis and degradation [102]. This is performed through control of the proteosome in nerve cells [103]. IPA analysis shows influences on both the mTOR signaling pathway and the proteosome network through pretreatment of THC.

The final metabolite of THC metabolism is THC-COOH. Pretreatment with this drug reduced S100B secretion and expression, as well as GFAP expression (Figure 1F,G). The pathway that was most affected by THC-COOH was IL6 signaling. IL6 is known to be elevated in activated EGCs [104]. Our data show downregulation of TGF-beta activated kinase 1 (*map3k7*) binding protein 1 (TAB1) and upregulation of TNF alpha induced protein 6 (*tnfaip6*) (−logFC −2.2 and 2, respectively). TAB1 is involved in IL6 activation through activation of IL-1 and NFƙB signaling [105]. Activation of TNFAIP6 inhibits IL-6 secretion in lung cells [106]. Together, our results indicate that preincubation with THC-COOH could reduce IL-6 secretion by EGCs.

## 4. Materials and Methods

### 4.1. Cell Culture

#### 4.1.1. J774A1 Murine Macrophages

J774A1 macrophages were purchased from the American type culture collection (ATCC, Manassas, VA, USA). Cells were cultured in 75 mm^2^ flasks with Dulbecco’s modified Eagle’s medium (DMEM) (Sigma-Aldrich, Saint Louis, MO, USA) supplemented with 10% fetal bovine serum (Biological industries, Kibbutz Beit-Haemek, Israel), 1% penicillin–streptomycin (Biological industries), and 2.5 mL sodium pyruvate (Biological industries) until they reached 70% confluency at 37 °C under 5% CO_2_.

#### 4.1.2. Enteric Glial Cells

Enteric glial cell lines (EGC/PK060399egfr) were purchased from the American type culture collection (ATCC). Cells were thawed and grown in 75 mm^2^ flasks to 70% confluence in DMEM medium containing 10% (Sigma-Aldrich) fetal bovine serum (FBS) (Biological industries) and 0.5% penicillin–streptomycin (Biological industries) at 37 °C under 5% CO_2_. Cells were trypsinized (using 0.25%) (Invitrogen, Carlsbad, CA, USA) and transferred every 2–3 days.

### 4.2. Chemicals

Pure THC was purchased from BOL pharma (Revadim, Israel). Purified CBD was obtained from Tikun Olam Ltd. (Tel Aviv-Yafo, Israel). Sparstolonin B (Ssnb) was purchased from Sigma-Aldrich. LPS THCA, THCV CBC, CBG, CBA, and THC-COOH were purchased at HPLC standard grade (Restek, Bellefonte, PA, USA). All purified or synthetic phytocannabinoids were dissolved in ethanol and later diluted with DMEM before addition to cells.

### 4.3. MTT 3-(4,5-Dimethylthiazol-2-yl)-2,5-diphenyltetrazolium Bromide

Cells were plated on 96-well plates at a concentration of 5 × 10^4^/0.2 mL/well and left to adhere for 2 h. The medium (DMEM D5796 Sigma-Aldrich) was replaced with media supplemented with different concentrations of test treatment. Cells were left in an incubator for 24 h (37 °C 5%CO_2_). The medium was replaced with 180 µL of clear medium (DMEM 01-053-1A Biological industries) supplemented with 20 µL solubilized MTT with a final concentration of 0.5 mg/mL (Sigma Aldrich) for 2 h. After removal of MTT, 100 µL of dimethyl-sulfoxide (DMSO) was added and left on an orbital shaker for 20 min. Absorbance was measured in a spectrophotometer (Synergy H1, Agilent, CA, USA) at 550 nm.

### 4.4. In Vitro Treatments

The cells were plated at a concentration of ~1 × 10^6^ cells/mL and were pretreated with different concentrations of single phytocannabinoids and/or a mixture of phytocannabinoids for 1 h based on previous studies [107], after which LPS (*E. coli* 0111:B4, Sigma USA) was added for an additional 24 h. After treatment, the medium was removed for ELISA analysis (see below), and RNA/proteins were extracted from the respective cells.

### 4.5. Enzyme-Linked Immunosorbent Assay (ELISA)

The cell’s growth medium was assayed for TNFα using ELISA, according to the manufacturer’s instructions (Peprotech, Cranbury, NJ, USA), or for S100B, using a Simple-Step ELISA kit according to the manufacturer’s instructions (Abcam, Waltham, MA, USA).

### 4.6. RNA Extraction and cDNA Synthesis

RNA was extracted using TRI reagent (Sigma GmBH, Mannheim, Germany) in combination with a PureLink column-based kit (Thermo Fisher, Waltham, MA, USA). RNA was quantified using Nanodrop 2000 (Thermo Fisher, Waltham, MA, USA). A total of 1.5 µg of RNA was used for synthesis of cDNA, using a qScript cDNA Synthesis Kit (Quanta Bio, Beverly, MA, USA).

### 4.7. Quantitative Reverse Transcription PCR (RT-qPCR)

Real time qPCR was preformed using fast SYBR green master mix (Applied Biosystems, Foster City, CA, USA) on a Quant studio 1 machine (Applied Biosystems). For normalization of gene expression in all reactions, we used the PPIA gene for TNFα gene normalization, and the GAPDH gene for S100B and GFAP genes normalization. Expression was quantified using the in-run standard curve method. Primers for relative gene expression are depicted in Appendix A.

### 4.8. RNA Sequencing Protocol and Computational Pipeline

For library construction and sequencing, total RNA was extracted, as described above. RNA-seq analysis was executed by the Crown Genomics institute of the Nancy and Stephen Grand Israel National Center for Personalized Medicine, Weizmann Institute of Science. A bulk adaptation of the MARS-Seq protocol [108,109] was used to generate RNA-Seq libraries for expression profiling. For sequence data analysis, assembly and annotation were performed, as described previously [110,111]. Differential analysis was performed using the DESeq2 package (1.26.0) [112] with the betaPrior, cooks cutoff, and independent filtering parameters set to False. Raw *p* values were adjusted for multiple testing using the procedure of Benjamini and Hochberg. Pipeline was run using snakemake [113]. DEGs were determined by a *p*-adj of <0.05, absolute fold changes > 1.6, and max raw counts > 10. For bioinformatics analysis, PCA, Hierarchical clustering, and K-Means clustering were performed. Standardized, log2 normalized counts were used for the clustering analysis. Clustering analysis was performed with R version 3.6.1. Data were submitted to GEO [114] accession GSE240225. DEGs, heatmaps, canonical pathways, and graphical networks were analyzed using Ingenuity Pathways Analysis (Ingenuity^®^ Systems version 90348151, www.ingenuity.com).

### 4.9. Statistics

All statistics were performed with JMP pro 14 (SAS institute Inc., ver 11, Cary, NC, USA, 1989–2019) or GraphPad prism (version 8 GraphPad Software, San Diego, CA, USA, www.graphpad.com). Unless otherwise stated, data are expressed as mean ± SE. Comparison between means of more than two groups were analyzed using means ANOVA and Tukey HSD.

## 5. Conclusions

Our results show that, while four different cannabinoids (CBC, CBG, THCA, and THC-COOH) reduced inflammation markers in EGCs, between all nine selected pure phytocannabinoids tested, only THC at low concentrations demonstrated significant reduction in TNFα secretion in J774A1 murine macrophages and EGC. This is in accordance with the known biphasic effect of THC, and it points to the advantage of using low doses of THC. Additionally, pretreatment of EGCs with low concentrations of THC markedly reduced S100B secretion and expression as well as GFAP expression. Taken together, these results show a significant immunosuppressive effect exerted by low doses of THC in our in vitro model. RNA-seq analyses and Ingenuity Pathways Analysis show that THC treatment affected the mTOR and RAR signaling pathway, while THC-COOH mainly affected the IL6 signaling pathway. The strength of the present study is that the putative anti-inflammatory effects of nine pure cannabinoids were analyzed in macrophages and enteric glial cells triggered to undergo inflammation in vitro with LPS tests that allowed us to select the most effective cannabinoid. The weakness of the present study is that we did not reflect these effects in in vivo studies, in which we hope to find similar effects in near future.

## Figures and Tables

**Figure 1 ijms-24-14628-f001:**
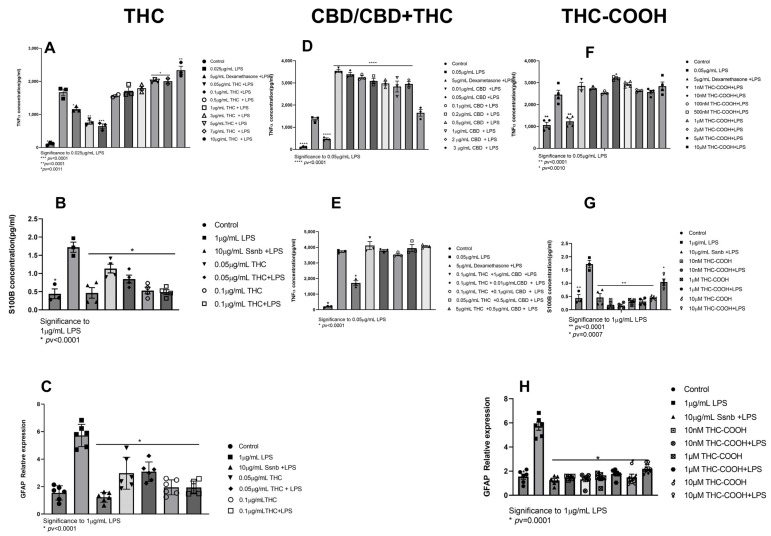
Pretreatment of J774A1 and EGC with THC (**A**–**C**), CBD (**D**,**E**), and THC-COOH (**F**–**H**) for 1 h and then incubation with LPS 0.05 μg/mL and 1 μg/mL for J774A1 and EGC, respectively, for an additional 24 h. Control represents J774A1/EGCs without treatment, 0.05/1 μg/mL LPS represents the positive control for J774A1/EGCs, and 5 μg/mL Dexamethasone and 10 μg/mL Ssnb treatment represents the negative control for J774A1 or EGC, respectively. (**A**) Secretion of TNFα by J774A1 cells treated with 0.05–10 µg/mL THC and 0.05 μg/mL LPS. (**B**) Secretion of S100B from EGCs pretreated with 0.05 and 0.1 μg/mL THC with and w/o 1 μg/mL LPS. (**C**) Expression of GFAP from EGCs pretreated with 0.05 and 0.1 μg/mL THC with and w/o 1 μg/mL LPS. (**D**) TNFα secretion from J774A1 pretreated with 0.01–3 μg/mL CBD with 0.05 μg/mL LPS. (**E**) TNFα secretion from J774A1 pretreated with a combination of THC + CBD and 0.05 μg/mL LPS. (**F**) TNFα secretion from J774A1 pretreated with 1 nM–10 μM THC-COOH with 0.05 μg/mL LPS. (**G**) Secretion of S100B from EGCs pretreated with 1 nM–10 μM THC-COOH with 0.05 μg/mL LPS. (**H**) Expression of GFAP from EGCs pretreated with 1 nM–10 μM THC-COOH with 0.05 μg/mL LPS. n = 3 for all J774A1 TNFα secretion, N = 4 for ECG S100B secretion, N = 6 for GFAP expression. All samples were compared to positive LPS control using Dunnett’s multiple comparison test.

**Figure 2 ijms-24-14628-f002:**
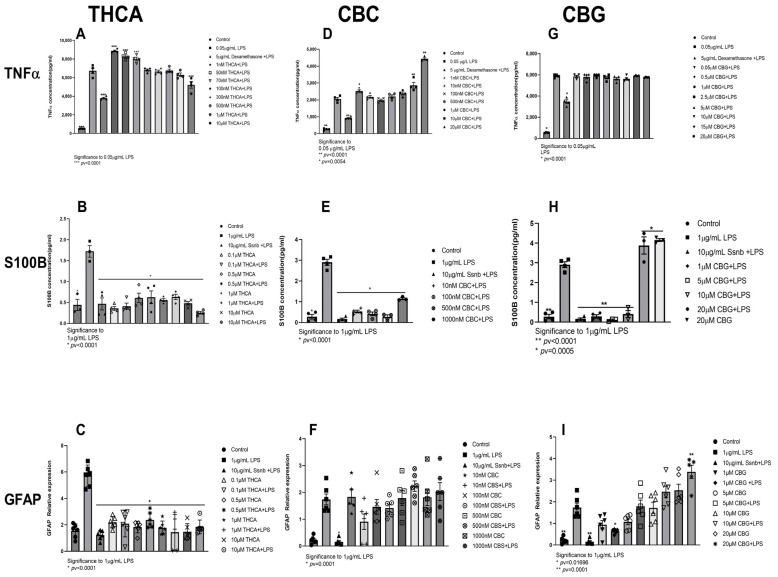
Pretreatment of J774A1 and EGC with THCA (**A**–**C**), CBC (**D**–**F**), and CBG (**G**–**I**) for 1 h and then incubation with LPS 0.05 µg/mL and 1 µg/mL for J774A1 and EGC for 4 h. Control represents J774A1/EGCs without treatment, 0.05/1 μg/mL LPS represents the positive control for J774A1/EGCs and 5 μg/mL Dexamethasone and 10 µg/mL Ssnb treatment represents the negative control for J774A1 or EGC, respectively. (**A**) Secretion of TNFα by J774A1 cells treated with 1 nM–10 µM THCA and 0.05 µg/mL LPS. (**B**) Secretion of S100B from EGCs pretreated with 1 nM–10 µM THCA with and w/o 1 µg/mL LPS. (**C**) Expression of GFAP from EGCs pretreated 1 nM–10 µM THCA and 0.05 µg/mL with and w/o 1 µg/mL LPS. (**D**) TNFα secretion from J774A1 pretreated with 1 nM–20 µM CBC with 0.05 µg/mL LPS. (**E**) Secretion of S100B from EGCs pretreated with a 1 nM–20 µM CBC with 1 µg/mL LPS. (**F**) expression of GFAP from EGCs pretreated with 1 nM–20 µM CBC with 1 µg/mL LPS. (**G**) TNFα secretion from J774A1 pretreated with 0.05–20 µM CBG with 0.05 µg/mL LPS. (**H**) Secretion of S100B from EGCs pretreated with a 1–20 µM CBG with 1 µg/mL LPS. (**I**) Expression of GFAP from EGCs pretreated with 1–20 µM CBG with or w/o 1 µg/mL LPS, N = 3 for all J774A1 TNFα secretion, n = 4 for ECG S100B secretion, N = 6 for GFAP expression. All samples were compared to positive LPS control using Dunnett’s multiple comparison test.

**Figure 3 ijms-24-14628-f003:**
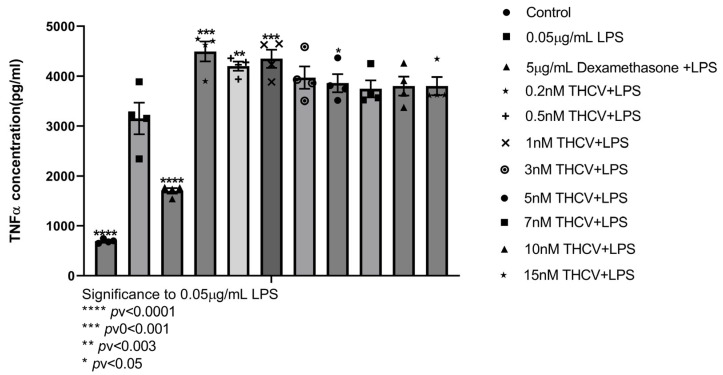
Pretreatment of J77A1 macrophages with increasing concentrations. J77A1 cells were pretreated with 0.2–15 nM THCV for 1 h and then incubated with 0.05 µg/L of LPS for 4 h. Control represents J774A1 cells without treatment, 0.05 μg/mL LPS represents the positive control, and 5 μg/mL Dexamethasone treatment represents the negative control. n = 4. All samples were compared to positive LPS control using Dunnett’s multiple comparison test.

**Figure 4 ijms-24-14628-f004:**
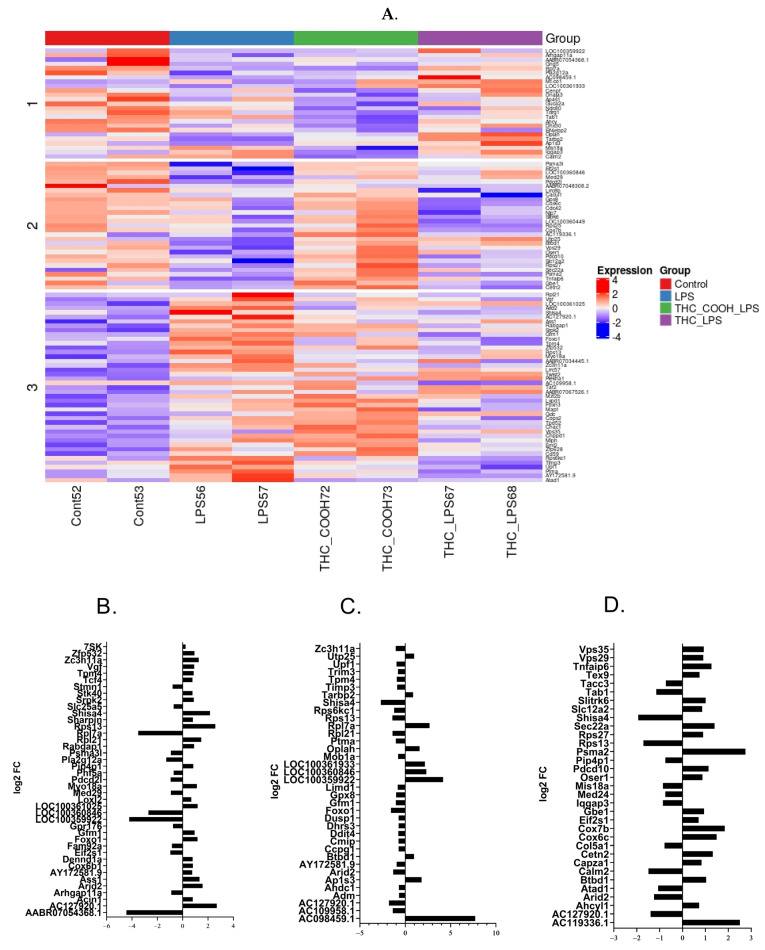
Differentially expressed genes (DEG) in enteric glial cells (EGC): (**A**): Differential expression of all groups: (**B**) Non-treated cell compared to cells treated with LPS 1 µg/mL for 24 h. (**C**): 1 h pretreatment with 0.1 µg/mL THC before 24 h of LPS compared to only LPS 1 µg/mL. (**D**): 1 h pretreatment of 10 nM of THC-COOH before 24 h of LP compared to only LPS 1 µg/mL. n = 2.

**Figure 5 ijms-24-14628-f005:**
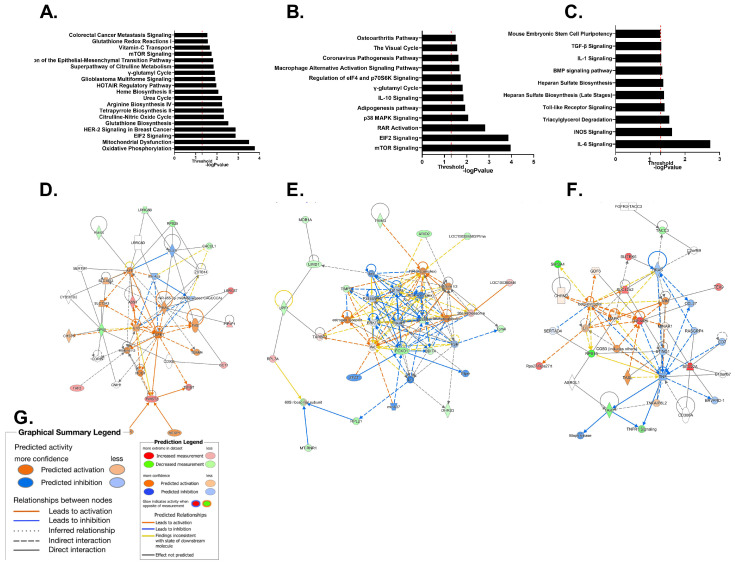
EGCs were either untreated or incubated with 1 µg/mL of LPS (**A**,**D**). In treatment groups, cells were pretreated for 1 h with 0.1 µg/mL of THC (**B**,**E**) or 10 nM of THC-COOH (**C**,**F**). IPA was conducted on three experimental groups. Pathway enrichment (**A**–**C**), networks showing most hits (**D**–**F**). Pathway analysis of (**A**) LPS vs untreated cells, (**B**) Cells pretreated for 1 h with 0.1 µg/mL of THC, (**C**) cells pretreated for 1 h with 10 nM of THC-COOH. Threshold 1.3 −log*p*v. (**D**) Cell cycle and cell death network affected by addition of LPS to untreated cells. (**E**) Proteosome network effected by pretreatment of THC. (**F**) Cell cycle and cell death network affected by pretreatment of cells with THC-COOH. (**G**) Legend for network. n = 2.

## Data Availability

Sequencing data is available at: https://www.ncbi.nlm.nih.gov/geo/query/acc.cgi?acc=GSE240225.

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
