# Peer review of "Phytocannabinoids Reduce Inflammation of Primed Macrophages and Enteric Glial Cells: An In Vitro Study"

_ijms, 2023, doi:10.3390/ijms241914628_

Round 1

Reviewer 1 Report

The manuscript "Phytocannabinoids reduce inflammation of primed macrophages and enteric glial cells" by Gal Cohen et al. is an experimental study where the anti-inflammatory properties of nine selected pure cannabinoids are analyzed in J774A1 macrophage cells and enteric glial cells. In general, the work is interesting (although it has already been evaluated in this context) and the experimental design is adequate. However, an in vivo study in experimental models would be necessary to be considered as a publication.

Author Response

The manuscript "Phytocannabinoids reduce inflammation of primed macrophages and enteric glial cells" by Gal Cohen et al. is an experimental study where the anti-inflammatory properties of nine selected pure cannabinoids are analyzed in J774A1 macrophage cells and enteric glial cells. In general, the work is interesting (although it has already been evaluated in this context) and the experimental design is adequate. However, an in vivo study in experimental models would be necessary to be considered as a publication.

We thank the reviewer for his comments. We give our response to each of the reviewer queries (our answers are in red, the reviewer queries in black).

The concept behind our study is to examine a wide array of cannabinoids in typical cells residing in the intestine and directly involved in inflammation such as macrophages and EGCs. Regarding the response of enteric glial cells (EGCs) to phytocannabinoids we believe that there lies our novelty. In this context some studies have been done in enteric neurons however EGCs have received much less attention. An in vivo study is out of the scope of this research.

Reviewer 2 Report

Greetings the work is very interesting and it would be convenient to see what happens in integro model where the regulation and physiological balance that adjusts an integro organism substantially modifies the responses, as well as the responses due to cross communication in the signaling pathways as well as the role in the inflammatory phenomenon to understand the potentiality of the study.
Some additional comments regarding the review of the paper: Phytocannabinoids reduce inflammation of primed macrophages and enteric glial cells.

  1. The study is important and addresses one of the potentialities of phytocannabinoids that has not been fully described. Although the study is in invitro, it provides information that may be relevant at the moment of treating an enteric organism and that may well be the basis for future studies and be able to consider what may happen in an enteric organism.  
  2. However, it addresses the effect of nine phytocannabinoids and the one that presents effect is THC, I suggest changing the title since it does not effectively express the finding and the conclusion it describes; the current title states that all phytocannabinoids present the effect and it is not so in the results therefore it is not traceable to the results and the conclusion.  
  3. In conclusion it should include the effect presented by CBC that argues in results since it only focuses on TCH and comment on the null effect by other phytocannabinoids.  
  4. It should include the figures with the most outstanding results and the other results can be included in a table and/or described in the text.  
  5. In the bibliography, the references should be updated, 80% of them are older than 5 years.

Author Response

Greetings the work is very interesting and it would be convenient to see what happens in integro model where the regulation and physiological balance that adjusts an integro organism substantially modifies the responses, as well as the responses due to cross communication in the signaling pathways as well as the role in the inflammatory phenomenon to understand the potentiality of the study.
Some additional comments regarding the review of the paper: Phytocannabinoids reduce inflammation of primed macrophages and enteric glial cells.

We thank the reviewer for the important comments, unfortunately for the present study in-vivo experiments are outside the scope of this research.  We give our response to each of the reviewer queries (our answers are in red, the reviewer queries in black).

  1. The study is important and addresses one of the potentialities of phytocannabinoids that has not been fully described. Although the study is in invitro, it provides information that may be relevant at the moment of treating an enteric organism and that may well be the basis for future studies and be able to consider what may happen in an enteric organism.

We completely agree with the reviewer that our study provides important information that may be relevant to treat in vivo organisms and be the basis for important future studies.

  1. However, it addresses the effect of nine phytocannabinoids and the one that presents effect is THC, I suggest changing the title since it does not effectively express the finding and the conclusion it describes; the current title states that all phytocannabinoids present the effect and it is not so in the results therefore it is not traceable to the results and the conclusion.

It is correct that only THC exerted an anti-inflammatory effect on both J77A1 and EGCs cells, however other cannabinoids also reduced inflammation in EGC. THC-COOH reduced both S100B and GFAP. So did THCA, CBC and CBG. Thus, we think that phytocannabinoids is still relevant in the title and discussion.

  1. In conclusion it should include the effect presented by CBC that argues in results since it only focuses on TCH and comment on the null effect by other phytocannabinoids.  

While CBC did show an interesting response, it did not significantly reduce TNFα secretion in J77A1 cells. The comment is important as other cannabinoids did affected EGC and the conclusion was corrected according to the reviewer’s comment.

  1. It should include the figures with the most outstanding results and the other results can be included in a table and/or described in the text.  

The current presentation of the data exemplifies the effect in both J77A1 cells and EGCs, which is a novel approach. This presentation enables the reader to easily compare the effect of each cannabinoid on both types of cells for all types of cannabinoids we used. 

  1. In the bibliography, the references should be updated, 80% of them are older than 5 years.

We thank the reviewer for the comment, references were updated.

Reviewer 3 Report

Acute inflammatory responses associated with enteric macrophages and glial cells are prominent features of inflammatory bowel diseases. In the present study, the authors have evaluated the potential of several phytocannabinoids to prevent or limit inflammatory responses triggered by lipopolysaccharide (LPS) in J774A1 macrophage cells and enteric glial cells (EGCs) in culture. Their data indicate that pretreatment of both cell types with Delta (9)-tetrahydrocannabinol (THC) at low concentrations significantly inhibited subsequent inflammatory responses to LPS, with the effects being most striking with ECGs. Comparative transcriptomic analysis of THC+LPS- and LPS treated-EGCs revealed that twenty-six genes were differentially expressed (10 upregulated and 16 downregulated) because of pretreatment with THC. Further analysis indicated that THC treatment affected the mTOR and RAR signalling systems.

This thorough and robust in vitro study demonstrates a potential role for phyto-cannabinoids, notably THC, in therapies for inflammatory bowel disease. However, the studies use THC as a prophylactic, which could be helpful for patients in remission. An important question would be whether it will be as effective for patients already in a flare-up. Is there any evidence of THC acting therapeutically with the presently used macrophages and enteric glial cells or in the literature with other cell types? Comment on therapeutic use in discussion.

Although there are separate results and discussion sections, it is difficult to really distinguish between them because they are very intermixed. The manuscript would have more impact if it focussed on the best phytocannabinoid identified and its known properties, demonstrated effects, strengths, and limitations.

THC was effective at low concentrations but ineffective or even deleterious at higher, apparently non-toxic levels. Are any of the reasons for this bell-shaped response known or speculated about? Could THC accumulate in cells over time and thereby lose efficacy?

Ln 45-47         This sentence is a better fit in paragraph 1.

Ln 49-50         ‘protect the body from inflammatory or infection’. Inflammation?

Ln 66-70         Transfer up to Ln 47 after [8].

Ln 71-72         ‘The gastrointestinal tract includes an extensive intrinsic nervous system termed the enteric nervous system (ENS), differing from all other peripheral organs. Alternatively, 'The gastrointestinal tract differs from all other peripheral organs in that it includes an extensive enteric nervous system (ENS)’.

Ln 160             ‘mirrored'. Mirror?    

Ln 274             ‘Pretreatment of THC before LPS’. ‘Pretreatment with THC before LPS.' Check the whole text for this misplacement.

Ln 311-312     ‘THC affected fewer pathways compared to THC above the threshold of 1.3 (Fig 5 b and Fig c). What is compared with what?

Ln 350-356     This should be in the results section.

Ln 365             switch ‘of’ to ‘with’

Ln 390             ‘prevent the transfer of EGC to reactive gliosis’. ‘prevent induction of reactive gliosis in EGCs’.

Ln 389-391     Expand this sentence to explain why the findings with colonocytes and C. sativa are consistent with the present data.

Minor editing of English is required.

Author Response

Acute inflammatory responses associated with enteric macrophages and glial cells are prominent features of inflammatory bowel diseases. In the present study, the authors have evaluated the potential of several phytocannabinoids to prevent or limit inflammatory responses triggered by lipopolysaccharide (LPS) in J774A1 macrophage cells and enteric glial cells (EGCs) in culture. Their data indicate that pretreatment of both cell types with Delta (9)-tetrahydrocannabinol (THC) at low concentrations significantly inhibited subsequent inflammatory responses to LPS, with the effects being most striking with ECGs. Comparative transcriptomic analysis of THC+LPS- and LPS treated-EGCs revealed that twenty-six genes were differentially expressed (10 upregulated and 16 downregulated) because of pretreatment with THC. Further analysis indicated that THC treatment affected the mTOR and RAR signalling systems.

This thorough and robust in vitro study demonstrates a potential role for phyto-cannabinoids, notably THC, in therapies for inflammatory bowel disease. However, the studies use THC as a prophylactic, which could be helpful for patients in remission. An important question would be whether it will be as effective for patients already in a flare-up. Is there any evidence of THC acting therapeutically with the presently used macrophages and enteric glial cells or in the literature with other cell types? Comment on therapeutic use in discussion.

We thank the reviewer for taking the time and thoroughly reviewing our manuscript (as appears from the valuable comments). We give our response to each of the reviewer queries (our answers are in red, the reviewer queries in black).

As for the questions about a therapeutic use we have added a few lines at the end of the discussion (lines 466-470). Please see answers to all comments below.

Although there are separate results and discussion sections, it is difficult to really distinguish between them because they are very intermixed. The manuscript would have more impact if it focussed on the best phytocannabinoid identified and its known properties, demonstrated effects, strengths, and limitations.

We thank the reviewer for this comment. The results section has been trimmed removing sentences better suited for the discussion. While only THC reduced inflammatory markers in both cell types other cannabinoids reduced inflammatory markers in EGCs either S100B secretion, S100B expression, GFAP expression, or a combination of them. Hence, we felt in necessary to provide the reader with an elaborate picture of the effects of the tested cannabinoids. 

THC was effective at low concentrations but ineffective or even deleterious at higher, apparently non-toxic levels. Are any of the reasons for this bell-shaped response known or speculated about? Could THC accumulate in cells over time and thereby lose efficacy?

Indeed, the biphasic effect of THC is an interesting phenomenon. We also saw it in another work we did that has just been just published (: https://www.mdpi.com/1422-0067/24/18/13797). This could be related to cellular tolerance but this is not fully understood [1-3].

Ln 45-47         This sentence is a better fit in paragraph 1.

Moved to a more adequate place (see lines 55-59)

Ln 49-50         ‘protect the body from inflammatory or infection’. Inflammation?

Corrected

Ln 66-70         Transfer up to Ln 47 after [8].

Moved

Ln 71-72         ‘The gastrointestinal tract includes an extensive intrinsic nervous system termed the enteric nervous system (ENS), differing from all other peripheral organs. Alternatively, 'The gastrointestinal tract differs from all other peripheral organs in that it includes an extensive enteric nervous system (ENS)’.

Changed.

Ln 160             ‘mirrored'. Mirror?   

Changed 

Ln 274             ‘Pretreatment of THC before LPS’. ‘Pretreatment with THC before LPS.' Check the whole text for this misplacement.

Corrected throughout the manuscript.

Ln 311-312     ‘THC affected fewer pathways compared to THC above the threshold of 1.3 (Fig 5 b and Fig c). What is compared with what?

Please not it is “Carboxylated THC affected fewer pathways compared to THC above the threshold of 1.3” (line 313)

Ln 350-356     This should be in the results section.

Moved to supplemental

Ln 365             switch ‘of’ to ‘with’

Corrected

Ln 390             ‘prevent the transfer of EGC to reactive gliosis’. ‘prevent induction of reactive gliosis in EGCs’.

Corrected

Ln 389-391     Expand this sentence to explain why the findings with colonocytes and C. sativa are consistent with the present data.

We expanded this sentence according to the reviewers comment.

  1. Katsidoni, V.; Kastellakis, A.; Panagis, G. Biphasic effects of Δ9-tetrahydrocannabinol on brain stimulation reward and motor activity. The international journal of neuropsychopharmacology 2013, 16, 2273-2284, doi:10.1017/s1461145713000709.
  2. Margulies, J.E.; Hammer, R.P. Δ9-Tetrahydrocannabinol alters cerebral metabolism in a biphasic, dose-dependent mannier in rat brain. European journal of pharmacology 1991, 202, 373-378, doi:https://doi.org/10.1016/0014-2999(91)90281-T.
  3. McKinney, D.L.; Cassidy, M.P.; Collier, L.M.; Martin, B.R.; Wiley, J.L.; Selley, D.E.; Sim-Selley, L.J. Dose-related differences in the regional pattern of cannabinoid receptor adaptation and in vivo tolerance development to delta9-tetrahydrocannabinol. The Journal of pharmacology and experimental therapeutics 2008, 324, 664-673, doi:10.1124/jpet.107.130328.

Reviewer 4 Report

The article of Cohen is well written and the data are well presented. 

My only issue is regarding the Discussion. I think that at the end of Discussion the obtained data should be compared with 2 or 3 similar studies in the same field. The authors can check on PubMED regarding the article more appropriate and compared them with the present study.

English is fine

Author Response

The article of Cohen is well written and the data are well presented. 

We thank the reviewer for the comments. We give our response to the reviewer queries (our answers are in red, the reviewer query in black).

My only issue is regarding the Discussion. I think that at the end of Discussion the obtained data should be compared with 2 or 3 similar studies in the same field. The authors can check on PubMED regarding the article more appropriate and compared them with the present study.

We thank the reviewer for his comment. Please see changes in lines 355-359 and in lines 375-377.

Round 2

Reviewer 1 Report

Although the authors have incorporated the reviewers' suggestions and in this way have improved the manuscript, I maintain that for a journal of this metric and quality it is necessary to test the results obtained in an in vivo model and thus ensure that it has an anti - inflammatory effect. In these experiments they have concluded that certain cannabinoids can modulate the expression of pro-inflammatory mediators (although without homogeneity in the results), but to verify that they have an anti-inflammatory effect it must be carried out in vivo. Taking into account that the manuscript is also being related to inflammatory bowel disease, with a clear chronic component. 

Reviewer 3 Report

All issues raised have been addressed in a satisfactory manner.
